# Msh Pilus Mutations Increase the Ability of a Free-Living Bacterium to Colonize a Piscine Host

**DOI:** 10.3390/genes12020127

**Published:** 2021-01-20

**Authors:** Jarrett F. Lebov, Brendan J. M. Bohannan

**Affiliations:** 1Institute for Genome Sciences, University of Maryland School of Medicine, Baltimore, MD 21201, USA; 2Department of Biology, Institute of Ecology and Evolution, University of Oregon, Eugene, OR 97403-5289, USA; bohannan@uoregon.edu

**Keywords:** host–microbe, zebrafish, shewanella, symbiosis, microbial evolution, msh pilus, adaptation, biofilms, fitness, colonization

## Abstract

Symbioses between animals and bacteria are ubiquitous. To better understand these relationships, it is essential to unravel how bacteria evolve to colonize hosts. Previously, we serially passaged the free-living bacterium, *Shewanella oneidensis*, through the digestive tracts of germ-free larval zebrafish (*Danio rerio*) to uncover the evolutionary changes involved in the initiation of a novel symbiosis with a vertebrate host. After 20 passages, we discovered an adaptive missense mutation in the *mshL* gene of the *msh* pilus operon, which improved host colonization, increased swimming motility, and reduced surface adhesion. In the present study, we determined that this mutation was a loss-of-function mutation and found that it improved zebrafish colonization by augmenting *S. oneidensis* representation in the water column outside larvae through a reduced association with environmental surfaces. Additionally, we found that strains containing the *mshL* mutation were able to immigrate into host digestive tracts at higher rates per capita. However, mutant and evolved strains exhibited no evidence of a competitive advantage after colonizing hosts. Our results demonstrate that bacterial behaviors outside the host can play a dominant role in facilitating the onset of novel host associations.

## 1. Introduction

Throughout evolutionary history, bacteria have routinely formed novel symbioses with animal hosts [1,2]. Some particularly well studied examples of these symbiotic interactions take place in the vertebrate digestive tract, which harbors dense bacterial communities whose constituents can impact a wide variety of biological processes in the hosts they inhabit [3,4,5,6]. Despite numerous studies aimed at disentangling host–microbe symbioses, much remains unknown about how bacteria evolve novel symbioses within the vertebrate gut, or which genomic features and corresponding traits enable host colonization. Resolving this information would provide invaluable insights into the determinants of vertebrate gut community membership and function. 

To this end, in a prior study we took an experimental evolution approach to elucidate which traits might enable a transition to host association by evolving a bacterium with no prior known host association to become better at colonizing a model vertebrate, the zebrafish *Danio rerio* [7]. For our experiment, we used the well-studied free-living bacterial species *Shewanella oneidensis,* strain MR-1, and cycled replicate populations through the intestines of germ free (GF) larval zebrafish. After 20 passages, we determined that a missense mutation in the *mshL* gene of MR-1’s mannose sensitive hemagglutinin (msh) pilus operon was capable of explaining a large portion of the adaptive gains exhibited by an evolved isolate containing this mutation. Interestingly, pili have been shown to play important roles in host colonization [8,9,10,11], and, in *Vibrio* strains, the msh pilus in particular has been shown to influence adhesion to mammalian intestinal epithelial cell lines [12], suggesting that the expression of these pili may be beneficial for host colonization. Alternatively, constitutive expression of the msh pilus in *V. cholerae* prevented colonization of infant mice when co-inoculated with secretory immunoglobulin A [13], implying that proper regulation of the msh pilus can be critical for bacterial establishment within vertebrate digestive tracts.

In our system, consisting of groups of larval zebrafish cohoused in culture flasks, it was unclear how a mutation in the msh pilus operon might improve the ability of MR-1 cells to colonize larval intestines. To access the larval gut, potential bacterial colonists added to the host’s habitat must be able to encounter a zebrafish larva and translocate from the external aqueous environment into the larval digestive tract. Once inside the gut, to be transmitted from one passage to the next, establishing populations must subsist until they are sampled at the end of a given passage. This process necessitates survival in a distinct set of environments in which expressing a pilus could be advantageous or deleterious. For example, pili are known for their involvement in aggregative behaviors [14,15,16,17,18,19] and could cause bacteria to associate with the surface of the flask in which they are housed or with particulates within the aqueous environment external to the fish. This type of sequestration could prevent bacteria from dispersing and accessing larvae swimming in their midst. Alternatively, these same behaviors could foster successful colonization upon encountering a zebrafish larva by enabling bacteria to adhere to the convoluted epithelial tissues that make up the intestine [11,12,20,21,22] or allowing bacteria to aggregate into clumps that shield internal members from abrupt chemical gradients and patrolling innate immune cells commonly found in vertebrate digestive tracts [23,24,25,26,27]. 

If these sorts of interactions were taking place, they should manifest themselves in some facet of the colonization process: augmentation in the water column to the point where bacteria–larvae encounters are increased in frequency, immigration into the larval gut, and survival in the gut. Fortunately, our zebrafish system allows us to independently assess which of these processes contribute to host colonization. In a prior genomic analysis where we compared the genomes of passaged and unpassaged MR-1 strains, we discovered an adaptive missense mutation in the *mshL* gene, which altered the coding of a threonine to a proline at MshL’s 300th amino acid position (MshL-T300P) [7]. Here, we determined this mutation to be a loss-of-function mutation which was linked to augmented representation in the water column and resulted in an increased likelihood of larval encounters. This tendency was caused by both an improved relative growth rate in the larval medium, as well as a reduced association with flask surfaces. Additionally, we found that the MshL-T300P mutation enhanced per capita immigration into larval intestines. Together, these traits allowed evolved isolates to colonize larval zebrafish digestive tracts in greater numbers than unpassaged MR-1 strains when the two types of strains were competing. These results demonstrate that the initial steps during a microbe’s transition to host-association may be driven primarily by mutations that alter its ability to access a host-based habitat, rather than mutations that increase fitness once a microbe has colonized a host. 

## 2. Materials and Methods

*Zebrafish husbandry:* To maintain animal research specimens under ethical conditions, we followed the strict guidelines laid out in the University of Oregon Institutional Animal Care and Use Committee (IACUC) protocol 15-98. Larval zebrafish were derived germ free per the considerations detailed in [28]. Germ free larvae could not be fed due to challenges in maintaining their food supply under sterile conditions. Therefore, IACUC protocol 15-98 mandates that all germ-free larvae be sacrificed at seven days post fertilization to prevent starvation. Specifics related to larval gut dissections can be found in [7]. Briefly, after euthanasia with tricaine (Western Chemical, Ferndale, Washington, DC, USA), larvae were mounted on glass slides coated with 3% methylcellulose to reduce specimen shifting during dissection. The process of dissecting larval guts can be found in [29]. Dissected guts were bullet-blended (Next Advance, Troy, New York, NY, USA) for 60 s at power four. Homogenized guts were dilution plated on tryptic soy agar to quantify bacterial contents by colony forming units. 

*Bacterial strains:* Karen Guillemin’s laboratory at the University of Oregon provided our wild type, unpassaged *S. oneidensis* strain (wt MR-1). We followed the procedures detailed in [30] to conduct all genetic modifications of wt MR-1. Briefly, Tn7-mediated insertion was used to incorporate neutral dTomato and green fluorescent protein gene tags. Whole gene deletion constructs were generated through splicing by overlap extension, while point mutation constructs were generated by amplification from evolved isolate genomes. In either case, constructs were incorporated into pAX2 vectors that were transformed into *Escherichia coli* DH5α, harvested, and electroporated into *E. coli* SM10. *E. coli* SM10 was then used to deliver the pAX2 vector into MR-1 cells by diparental mating. Merodiploid intermediates were streaked on tryptic soy agar plates and screened for second recombination events. Incorporation of mutation constructs in wt MR-1 was confirmed by polymerase chain reaction (gene deletions) or DNA sequencing (point mutations). *E. coli* strains were cultured in LB broth at 37 °C, except during diparental mating with wt MR-1, where they were cultured in tryptic soy broth at 30 °C. All MR-1 derivatives were cultured in tryptic soy broth at 30 °C unless otherwise indicated. 

*Orthologous protein comparisons:* To find proteins with similar structural elements to wt MR-1 MshL, the full amino acid sequence of MshL was submitted to BLASTp (https://blast.ncbi.nlm.nih.gov/Blast.cgi?PAG=Proteins) using the Protein Data Bank (https://www.rcsb.org/) as the selected database. The accession with the highest identity to wt MR-1 MshL (6I1X) was then downloaded and aligned residues 209–479 were visualized with PyMOL (https://pymol.org/2/). BLASTp was then used to cultivate a set of 1247 putative orthologous amino acid sequences using a 31 amino acid stretch that spanned 15 amino acids on either side of MshL-T300. To acquire this many sequences, we altered the algorithm parameters to include a maximum of 10,000 target sequences, and then filtered the results for sequences that shared 70–90% identity with our MshL reference. These sequences were then downloaded and entered into COBALT (https://www.ncb.nlm.nih.gov/tools/cobalt/cobalt.cgi) for alignment. The resulting alignment file was entered into WebLogo 3 (http://weblogo.threeplusone.com/) to generate a depiction of the amino acid representation at each site. Amino acid percentages at sites that corresponded with MshL-T300 were calculated manually (Appendix A). 

*Whole experimental system competitions:* Whole system competitions were performed identically to the “Competition assays” described in [7]. Briefly, overnight cultures of competitor strains were subcultured, harvested at late log stage, and mixed 1:1. This mixture was used to inoculate flask containing groups of four days post fertilization (dpf) larval zebrafish, and inoculating ratios were confirmed by colony forming unit (CFU) counts. Strains were distinguished using fluorescence microscopy by the presence or absence of a neutral fluorescent tag. At 7 dpf, larval guts were dissected, homogenized, and plated to determine CFUs for each competitor. Competitive indices were then determined as follows: (1)$$\mathrm{Competitive}\;\mathrm{index}=\;\left(\frac{\mathrm{Competitor}1:\mathrm{Competitor}2_{\mathrm{guts}}}{\mathrm{Competitor}1:\mathrm{Competitor}2_{\mathrm{inoculum}}}\right)$$

Reported values were log transformed so that they were not lower bound by zero. Log-transformed values below zero indicate a disadvantage for Competitor 1, while values above zero indicate an advantage for Competitor 1. 

*Biofilm assay:* Biofilm assays were previously described in [7]. Briefly, biological replicate overnight tryptic soy broth (TSB) cultures were pelleted by centrifugation, normalized by resuspension in larvae conditioned medium (LCM; described below), and 150 µL samples were deposited into a 96-well polystyrene plate. After a 24 h incubation at 30 °C, the volume of the wells was removed, and each well was rinsed with embryonic medium (EM) and stained with crystal violet. Crystal violet was then solubilized from biofilms within each well with 100% dimethyl sulfoxide (DMSO), and the absorbance of each well was determined by spectrometer at 570 nm.

*Evolved mutation calling:* A detailed description for how isolates were collected from evolved populations and sequenced can be found in [7]. Briefly, Nextera XT DNA sequencing libraries were prepared for each isolate and sequenced on the Illumina HiSeq 4000 (Illumina, San Diego, California, USA). Breseq 0.31.0 was then run on evolved and ancestral isolates in consensus mode to determine evolved mutations [31]. 

*Colonization level over time:* Overnight TSB cultures of competing strains were diluted 1:100 in TSB and allowed to grow out to late log stage (4–5 h). Then, 500 µL of each competitor were mixed in a single 1.7 mL tube so that competitors were at an approximate 1:1 ratio. Competition mixtures were pelleted (7000 rcf for 5 min) and resuspended in 1 mL sterile EM. Resuspended competition mixtures were diluted 1:100, and 7.5 µL of these dilutions were used to inoculate GF larval flasks containing ~30 mL of EM and ~30–35 larvae at 4 dpf. At 0, 2, 4, and 8 h post inoculation, 100 µL of the EM containing the larvae were sampled and dilution plated to determine the CFU/mL for each competing strain, and then five larval guts were immediately dissected and their contents dilution plated (described in Whole experimental system competitions above) to determine the CFU/gut of each competing strain. 

*Larvae-conditioned media (LCM) competitions:* Overnight TSB cultures of competing strains were diluted 1:100 in TSB and allowed to grow out to late log stage (4–5 h). Then, 500 µL of each competitor were mixed in a single 1.7 mL tube so that competitors were at an approximate 1:1 ratio. Competition mixtures were pelleted (7000 rcf for 5 min) and resuspended in 1 mL sterile EM. Resuspended competition mixtures were diluted 1:100, and 7.5 µL of these dilutions were used to inoculate GF larval flasks where larvae had been removed at 4 dpf. Each of these GF larval flasks had contained ~15 mL of EM and ~15 larvae that had been conditioning the EM over the course of four days. Multiple GF larval flasks had been combined into a single sterile vessel, and the larvae were removed using a 1.5 mL bulb pipette, leaving behind only the LCM. ~15 mL of LCM was then aliquoted into fresh sterile polystyrene culture flasks. These flasks were inoculated as stated above and triplicate 100 µL LCM samples were taken per competition flask and dilution plated to establish an initial competition ratio (Competitor 1:Competitor 2) according to the CFUs counted for each strain. Each competition flask was incubated at 28 °C for three days (to mimic a single passage during our serial passage procedure) at which point 100 µL samples were again taken in triplicate and dilution plated to establish a final competition ratio (Competitor 1:Competitor 2) by CFU. Competitive indices were calculated by dividing the final CFU ratio of Competitor 1:Competitor 2 by the inoculation ratio. Follow-up shaking and stationary LCM competitions were performed over the course of 24 h, rather than 72 h (Appendix A). Final cell counts near flask surfaces were determined in MATLAB using previously described cell tracking software (https://pages.uoregon.edu/raghu/particle_tracking.html) to analyze images from three to five randomly chosen locations in focus near the bottom of each replicate flask. Fluorescence microscopy-based imaging (Nikon Eclipse Ti-e) was used to distinguish competitors tag with neutral dTomato and green fluorescent protein markers. Competitive indices near flask surfaces were calculated as above by dividing the image-based ratio of each competitor found near the flask surfaces by the inoculating ratio determined by CFUs. In shaking LCM competitions, flasks were shaken on a rotator at 180 rpm. 

*Immigration and* in vivo *growth assays:* Separate 5 mL overnight TSB cultures of competitor strains of interest were diluted 1:100 in TSB and allowed to grow out to late log stage (4–5 h). Separate 1 mL subculture samples from each competitor were then pelleted (7000 rcf for 5 min), and the pellets were resuspended in 1 mL sterile EM. Resuspensions for each competitor were then diluted 1:100, and 7.5 µL of each dilution were used to inoculate separate 15 mL LCM flasks (created as described in LCM competitions assays above). Each competitor was incubated in its own LCM flask at 28 °C for 12–15 h, at which point each competitor’s flask contents was combined into a single polystyrene petri dish such that the petri dish contained a 30 mL competition mixture. 30–35, 4 dpf larvae were then added to each competition mixture dish, and 100 μL samples were immediately taken in triplicate from each dish. These samples were dilution plated to establish an inoculating competition mixture (Competitor 1:Competitor 2). After a 40–60 min incubation period at 28 °C, during which time larvae were colonized by competing populations of MR-1 strains, the LCM was removed and replaced with 100 mL sterile EM for a total of three rinses. These media exchanges reduced the density of each *S. oneidensis* competitor in the EM to prevent further immigration (colonization threshold > 10^4^ CFU/mL; Appendix A). Ten larval guts were then dissected and plated, as described above in our whole experimental system competitions, to determine the founding population sizes for each competitor in each larval gut. Immigration indices were then calculated per dissected gut by dividing the founding competitor ratio in dissected guts (Competitor 1:Competitor 2) by the initial ratio larvae were exposed to. Following these gut dissections, EM exchanges were repeated every 2 h to ensure that bacterial loads in the EM were kept below 10^3^ CFU/mL, preventing additional immigration into the remaining larvae. The larvae that remained in the competition petri dishes after the first set gut dissections were incubated at 28 °C between sterile EM exchanges. Ten hours after initial gut dissections took place, 10 additional larval guts were dissected, and their contents were plated to establish a final mean competition ratio for a typical larval gut. Since the CFU/mL in the EM was maintained at a low level between bouts of dissection, any increase in the mean CFU/gut we observed between dissection bouts should have been affected primarily by the growth of *S. oneidensis* populations in vivo. Using the founding mean CFU/gut and the final mean CFU/gut that we observed for each competing strain, we calculated a per capita growth rate for each competing strain [32]:(2)$$r=\;\frac{\ln\left(\frac{{\mathrm{CFU}\;\mathrm{per}\;\mathrm{gut}}_{\mathrm{final}}}{{\mathrm{CFU}\;\mathrm{per}\;\mathrm{gut}}_{\mathrm{initial}}}\right)}{\mathrm{time}}$$
and then used our per capita growth rate to calculate an *in vivo* competitive index: (3)$$in\;vivo\;\mathrm{competitive}\;\mathrm{index}\;=\;\frac{r_{\mathrm{Competitor}\;1}}{r_{\mathrm{Competitor}\;2}}\;$$

In this way, our in vivo competitive indices compare the relative per capita growth rates we observed for each competitor. 

*Evolved mutation calling:* A detail description of the methods we used for evolved mutation calling was previously provided in [7]. Briefly, we obtained four isolates from six replicate evolved populations (24 isolates total). For single-end 150 base pair genomic library preparations and sequencing on the Illumina HiSeq 4000, we used the Promega Wizard^®^ Genomic DNA Purification Kit (Catalog #: A1120) to extract genomic DNA from each evolved isolate and wt MR-1. Evolved mutations, were identified using breseq 0.31.0 (in consensus mode) to separately compare each Illumina sequenced isolate to the annotated ancestral reference [31]. All gene annotations featured in Appendix A were determined by Prokka v1.12 [33], except for the mshH-Q gene annotations which were determined by RAST v 2.0 [34].

## 3. Results

### 3.1. Comparative Analysis Suggests MshL-T300P Is a Loss-of-Function Mutation

In our prior work, we engineered the MshL-T300P mutation in the wild type (wt) MR-1 genome. This mutation provided nearly a 10-fold improvement in larval colonization relative to wt MR-1, and it was associated with a reduction in biofilm formation, as well as an increase in swimming motility [7]. Given that the msh operon encodes a pilus which plays a crucial role in the initiation of biofilms [14,35], this result implies that the MshL-T300P mutation could be a loss-of-function mutation. To assess the likely effect of the MshL-T300P mutation, we conducted a BLAST search of wt MR-1’s MshL amino acid sequence in the Protein Data Bank (PDB; https://www.rcsb.org/) and compared it to the most closely related hit, *Aeromonas hydrophila* ExeD (PDB accession: 6I1X). ExeD has been shown to form a pentadecameric pore complex [36], which agrees well with the role others have predicted for MshL [15,37,38]. The BLAST algorithm was able to align a stretch of 299 amino acids at a sequence identity of 31%, which spanned position 300 in MshL. From this, we inferred that the wt MR-1 threonine residue at MshL position 300 was likely part of a β-sheet (Appendix A). Because threonine is much more likely to be found in β-sheets than proline according to Chou–Fasman secondary structure predictions [39], we suspected that the missense mutation we observed would be unlikely to be found in proteins that were homologous to MshL. To assess this possibility, we collected a set of 1247 proteins that shared 70–90% identity with a 31 amino acid stretch spanning the MshL-300 site. We found that threonine was present at this position in 79.3% of the peptide sequences we considered, and that proline was never observed at this site (Appendix A and Appendix A). Of the several other non-threonine amino acids found at sites corresponding to the wt MR-1 MshL-300 site, glutamate was the most represented at 4.8% (Appendix A). 

The lack of proline among sites aligning to the MshL-300 site and the Chou–Fasman rules (which suggest that proline is comparatively rare in β-sheets) led us to hypothesize that a proline residue at the MshL-300 site would alter the stability of MshL. To assess this likelihood, we used INPS-3D, a server that predicts the effect of nonsynonymous mutations on protein stability using PDB files and mutational information [40]. Because the structure of MshL has not been solved, we compared the effects of threonine, glutamate, and proline mutations on *A. hydrophila* ExeD stability. Mutating the ExeD alanine, which aligned to wt MR-1’s MshL-300 site, to either a threonine (ΔΔG: 0.16) or a glutamate (ΔΔG: −0.26) was predicted have a much smaller impact on ΔG than that of an alanine to proline mutation (ΔΔG: −1.08). This suggests that the MshL-T300P mutation we observed could have a substantial impact on MshL stability and the function of the pore it is predicted to form [15,37,38]. 

### 3.2. MshL-T300P Behaves Similarly to a MshL Deletion Mutant

To confirm that MshL-T300P was a loss-of-function mutation, we separately assayed the ability of a MshL deletion mutant (ΔMshL) to colonize larval digestive tracts and form biofilms. Given our previous finding that the MshL-T300P mutation improved colonization of larval guts and caused a reduction biofilm formation relative to wt MR-1, we hypothesized that, if the MshL-T300P mutation were a loss-of-function mutation, the MshL-T300P and ∆MshL mutants would perform similarly. We assessed the relative ability of ΔMshL mutants to colonize larvae via competition against wt MR-1 strains that were neutrally tagged with a fluorescent marker. For these competitions, we quantified the colony-forming units (CFUs) of each strain in our inoculum and in larval guts 72 h after exposure. We then calculated a competitive index which allowed us to quantify the ability of a competing strain to colonize larval digestive tracts relative to our unpassaged wt MR-1 strain (see “Whole experimental system competitions” in Section 2: Materials and Methods for details [32,41]). Log transformed index values above zero indicate a strain’s competitive advantage, while values below zero indicate a disadvantage. The results from this assay demonstrate that the ΔMshL mutant significantly outcompeted the wt MR-1 strain producing a log(competitive index) of 1.15 ± 0.68 (mean ± standard deviation) among 30 dissected larval guts from three separate flasks [42]. This competitive advantage was of similar magnitude to what we had previously observed for the MshL-T300P (0.84 ± 0.50 [7]). Additionally, in crystal violet staining assays, where higher crystal violet absorbance readings indicate more robust biofilms, the ∆MshL mutant exhibited an absorbance that was commensurate with that of the MshL-T300P mutant and an evolved isolate containing the MshL-T300P mutation (L3a) [7], but was reduced compared to what we observed for wt MR-1 (Figure 1) [42]. The evolved L3a isolate was obtained after 20 passages during our serial passage scheme, and contained the MshL-T300P mutation plus several other mutations that accumulated (Appendix A) [7]. Phenotypic similarities between the MshL-T300P mutant and a mutant where the entire *mshL* gene was deleted suggest that the MshL-T300P mutation hinders the normal function of the *mshL* gene. Furthermore, the reduced biofilm phenotype of both the MshL-T300P and ΔMshL mutants suggests that they may not be properly expressing type IV pili, which are crucial for biofilm formation in wt MR-1 [14,35]. This is consistent with a proline residue at the MshL-300 site disrupting normal protein folding and/or processing. 

### 3.3. Evolved Lineages Show No Improvements in Carrying Capacity In Vivo

After confirming the effect of the MshL-T300P mutation on MshL, we next sought to understand how this mutation improved host colonization. Previously, we had observed that a *Shewanella* species isolated from larval zebrafish digestive tracts was able to colonize GF larval guts in greater abundances than wt MR-1 [7], and this characteristic presented an intuitive metric by which gut-adapted strains might manifest improved fitness in our experimental system. Thus, we examined the bacterial abundances present in larval zebrafish guts at the end of competitive fitness assays, expecting that bacterial cell densities in the gut would be elevated when strains containing the MshL-T300P mutation were present. Our data show that neither the MshL-T300P mutant nor an evolved isolate which contained this mutation (L3a) exhibited cell densities above what was expected for wt MR-1 (Figure 2). These results suggest the MshL-T300P mutation does not increase carrying capacity within the gut [42].

### 3.4. L3a and T300P Have Similar In Vivo Growth Rates to the wt MR-1 Strain

Given that we did not observe differences in colonization abundances that could explain the fitness advantage we observed for the L3a and MshL-T300P strains in our system (Figure 2), we hypothesized that the MshL-T300P mutation must impart its advantage through other components of fitness. In our experimental system, in order for bacterial cells to be propagated from one passage to the next, populations must grow to a density where they can readily colonize larvae, traverse host filters to migrate from the aqueous environment into the larval digestive tract and grow within the digestive tract. Isolates that have increased their ability to perform any of these tasks relative to the ancestral strain should exhibit increased fitness via our competitive assay [42].

We first suspected that our evolved populations might have improved in their ability to compete against the wt MR-1 strain in vivo once they had colonized larval guts. However, because our gut sampling scheme was destructive, we were unable to calculate growth rates for competing MR-1 populations within a single larval gut. Although others have quantified in vivo bacterial growth rates in a limited number of individual live larvae using light-sheet microscopy [43], it is not yet feasible to count bacteria in multiple living larvae simultaneously, and larvae do not produce fecal material that can be easily sampled for bacterial quantification. To overcome this limitation, we relied on methods that closely approximate population dynamics in a typical larval gut. We dissected the guts of groups of 10 larval zebrafish colonized with competition mixtures of L3a or MshL-T300P versus the wt MR-1 strain at two different time points. Ten larval guts were dissected at an initial time point to establish a mean founding population size for each competitor in a typical larval gut, and 10 additional larval guts were dissected 10 h later, after some growth in the gut had taken place, to establish a mean final population size in a typical gut. We estimated that bacterial populations in a typical larval gut underwent 3.5 ± 1.43 (mean ± sd) doublings during this 10 h window (Appendix A). Based on our examination of the initial colonization dynamics of a separate evolved population containing a MshL-T300P isolate while it was in competition with its wt MR-1 ancestor (evolved population 2; Appendix A), we determined that larval guts were not colonized at cell densities below 10^4^ CFU/mL (Appendix A). Therefore, we regularly exchanged the embryonic medium (EM) in our in vivo growth experiment to ensure that external bacterial cell densities were kept substantially below this threshold, so that cells detected in the gut at the end of our experiment were most likely the result of in vivo growth. At the conclusion of our in vivo growth experiment, we were able to calculate a relative fitness metric based on the per capita growth rates of each competitor (see methods for details [32]). We found that neither the L3a isolate nor the MshL-T300P mutant outcompeted wt MR-1 in vivo, and, surprisingly, it appears that the MshL-T300P isolate may even underperform the wt MR-1 strain (Figure 3) [42]. 

### 3.5. L3a and T300P Outcompete the wt MR-1 Strain in Larvae-Conditioned Medium

Because our adaptive strains could not outcompete our wt MR-1 strain in vivo, we next sought to investigate whether a portion of the L3a isolate’s fitness was based on its adaptation to the external environment. Thus, we conducted competition assays in larvae-conditioned medium (LCM) that was devoid of larval hosts. LCM was generated by deriving zebrafish embryos germ free in embryonic medium (EM), and incubating them for four days at 28 °C, during which time larvae hatched from their chorions and conditioned the EM. Larvae were then removed from the resulting LCM, leaving behind a growth-supporting medium that mimicked the nutrient profile experienced by evolving MR-1 strains during serial passage. In LCM competitions, competitive indices were calculated by dividing final mutant:wt MR-1 ratios by initial ratios after a 72 h incubation to simulate the duration of a single passage. We found that both the MshL-T300P mutant and the evolved L3a isolate were able to outcompete the wt MR-1 strain in this assay, implying that part of the colonization advantage of the evolved and mutant strains stemmed from their elevated densities in the water column (Figure 4A) [42]. 

For these assays, samples were taken by pipette from the water column to determine competitor ratios. Thus, the only way one strain could outcompete another is by increasing its representation specifically in the planktonic portion of the environment (where the larvae would normally reside), rather than near the surface of the flasks. Given our previous finding that the MshL-T300P mutant and the L3a isolate formed reduced biofilms compared to wt MR-1 [7], we wondered whether their more planktonic representation stemmed from their decreased ability to engage with surfaces. If one competitor was less able to adhere to flask surfaces, its cells should be more likely to escape back into the water column after contacting a flask surface. To address this possibility, we conducted additional LCM competitions in which we used fluorescence microscopy combined with an image-based cell-tracking algorithm to quantify the proportion of each competitor’s population that was flask-surface-associated or planktonic. In these assays, we considered cells that were in focus near the flask surface to be flask-associated. As in Figure 4A, we observed that L3a was able to outcompete wt MR-1 in the water column (Figure 4B). However, this trend was reversed when we quantified the competitive relationship near the flask surface where wt MR-1 outcompeted the L3a isolate, suggesting that the elevated occupancy of the L3a isolate in the water column stemmed from its reduced ability to adhere to flask surfaces (Figure 4B). 

To corroborate these findings, we conducted additional LCM assays in which competition flask contents were continuously mixed via shaking. We hypothesized that if the flask contents were thoroughly mixed, it would reduce the propensity for bacterial cells to adhere to flask surfaces, thereby causing both competitors to maintain more planktonic populations. Under these circumstances, we expected each strain to compete more equally. We found that shaking our LCM competitions produced approximately a 10-fold reduction in flask-associated populations compared to stationary conditions despite similar founding population sizes and similar numbers of total population doublings (flask associated + planktonic) after 24 h (Appendix A). We also observed that L3a’s competitive advantage in the water column was reduced under shaking conditions (Figure 4B, dark gray bars). Conversely, near the flask surface L3a had a higher mean competitive index under shaking conditions compared to stationary conditions (Figure 4B, white bars). In both types of LCM competitions (shaking and stationary), when we quantified the total population by combining the cells counted both in the water column and near the flasks surface, we found that L3a outcompeted the wt MR-1 strain. Together, these results demonstrate that not only was the L3a isolate less likely to be sequestered by flask surfaces than the wt MR-1 strain, but it was also capable of reaching larger total population sizes in LCM when both strains were competing. These results capture the exact behavior of an isolate which evolved during our serial passage experiment [7], and, given close agreement of the biofilm and LCM competition phenotypes between the L3a isolate and the MshL-T300P mutant, it is highly likely the results featured in Figure 4B stem from the MshL-T300P mutation. 

### 3.6. L3a and MshL-T300P Increase per Capita Immigration into the Gut

Although the evolved and mutant strains were able to outcompete the wt MR-1 strain based in part on their greater representation in the planktonic phase of our experimental setup, we wondered whether these strains could also immigrate into the gut more efficiently on a per capita basis. To assess this, we exposed GF larvae to competition mixtures containing strains that had first been independently cultured in LCM and then mixed together. We then calculated immigration indices which accounted for the starting ratio of each competitor and the ending ratio we observed in larval guts 1 h post exposure. For both the L3a and MshL-T300P mutant competitions, the mutant and evolved strain outcompeted wt MR-1 in terms of their per capita colonization of the larval gut (Figure 5). This result suggests that the MshL-T300P mutation can increase the rate at which MR-1 cells migrate from the aqueous external environment into the larval zebrafish gut [42]. 

## 4. Discussion

While our previous work uncovered an adaptive mutation which augmented swimming motility [7], here we established that this mutation was a loss-of-function mutation and pinpointed the primary modes by which this mutation increases colonization of the gut. We found that the MshL-T300P mutation decreased surface adhesion and provided a growth advantage in larvae-conditioned media, resulting in greater representation in the aqueous environment evolving MR-1 populations shared with larval zebrafish (Figure 4A,B). Further, perhaps stemming from their increased motility [7], MshL-T300P mutants also displayed higher levels of per capita immigration into the larval digestive tract (Figure 5). Surprisingly, this mutation provided no apparent improvement in fitness to MR-1 populations once they had colonized the gut (Figure 3). However, given that MshL-T300P mutants significantly outperformed wt MR-1 in accessing the gut environment, our findings suggest adaptations which alter dynamics external to the host can influence the propensity for bacteria to evolve host associations.

The increase in host colonization imparted the MshL-T300P mutation likely stems from the loss-of-function effect it has on the MshL protein. MshL assembles into an inner membrane pore complex through which the msh pilus extends [15,37,38,44]. If the function of MshL pore complex were compromised, which is a likely outcome given our comparative analysis (Appendix A), it could reduce the expression of the msh pilus on the surface of MR-1 cells. Given that others have found the msh pilus to play a crucial role in the initiation of MR-1 biofilms [14,35], this should result in attenuated biofilm formation, which is exactly what we observed for the MshL-T300P and ∆MshL mutants. Therefore, loss of MshL function resulting in diminished msh pilus expression presents a plausible physiological mechanism by which the MshL-T300P mutation enhances host colonization: bacterial cells that are unable to express pili are less likely to be sequestered in biofilms after contacting a flask surface, and would thus be more likely to escape back into the water column where they can encounter and colonize larval hosts. 

Given the colonization advantage imparted by the MshL-T300P mutation, it is interesting to consider whether this mutation could foster long-term host associations. If so, one might expect to find mutations which similarly inactivate msh pilus expression among other host-associated strains. However, in *Vibrio* strains, the msh pilus can play an important role as a host colonization factor [12,13,45,46], suggesting that alterations of its function could have detrimental effects on the longevity of host–microbe symbioses. Additionally, in our previous examination of a closely-related *Shewanella* zebrafish isolate, which also contained an *msh* pilus operon and exhibited competitive indices that were significantly higher than the MshL-T300P mutant, we did not find compelling evidence to suggest its *msh* pilus was inactive [7]. Thus, it remains possible that the MshL-T300P mutation could represent a temporary evolutionary change that improves colonization in the short-term, but which could be maladaptive for zebrafish colonization, relative to more fit genotypes, over longer time scales. Indeed, loss-of-function mutations are common among bacterial strains facing novel sets of conditions [47], and they may be purged from a population over time through competition with better adapted competitors [48,49]. In our system, MshL-T300P mutants could be outcompeted in a longer evolutionary experiment that yielded clones which maintained a similar ability to access larval hosts but, unlike the MshL-T300P mutant, also exhibited elevated fitness inside the host gut. Alternatively, if the MshL-T300P mutation constitutively represses biofilm formation, mutants that have this mutation might be outcompeted under more natural conditions in which microbial communities are present and contain members whose biofilm phenotypes are more nuanced. For example, multiple bacterial species have been shown to form biofilms which enable close associations with host tissues [12,50,51,52,53,54], and biofilms can protect bacteria from environmental stressors [55,56,57], enabling survival until they are able to colonize a host [26]. Dynamics such as these might explain why loss-of-function msh pilus mutations may be less likely to evolve among host-associated bacteria. Nonetheless, the fact that the MshL-T300P mutation increased the ability of a non-host-associated strain to colonize a zebrafish host suggests that its corresponding phenotypes could play an influential role during a bacterium’s transition to host-association. 

Less clear is how the MshL-T300P mutation might provide a growth advantage in LCM. In our LCM assays, if one competitor had reduced representation in the water column that was counteracted by its overrepresentation near the flask’s surface, we should be able to resolve such a discrepancy by quantifying both portions of its total population (planktonic + flask associated). We found that the total population size of an evolved isolate containing the MshL-T300P mutation (L3a) was larger than wt MR-1 after a period of competitive growth under both stationary and shaking conditions, suggesting faster growth under both conditions (Figure 4B). One possible explanation for this is that under both conditions, L3a cells appeared to be less prone to aggregation in biofilms and therefore may have been freer to forage for nutrients. Alternatively, compared to its wt MR-1 ancestor, the L3a isolate contained additional mutations which the MshL-T300P mutant did not. It is possible that some of these mutations contributed to the higher total population densities we observed for L3a. 

We also found that the MshL-T300P mutation was sufficient to recapitulate the per capita immigration advantage evolved isolate L3a had over wt MR-1. The exact mechanism for how the MshL-T300P mutation might increase transmission into the larval gut from the external aqueous environment remains mysterious, however it is tempting to ascribe some credit to the increase in swimming motility imparted by the MshL-T300P mutation we reported previously [7]. Faster swimming speeds could result in greater rates of per capita colonization by making chemotactic responses to larvae-produced gradients more efficacious, and they could also allow for more rapid diffusion throughout the experimental medium [58,59]. In either case, hypermotility would increase physical encounters between hosts and microbes, resulting in more frequent transmission into the host’s digestive tract compared to a less motile competitor. 

Interestingly, this finding is consistent with what Robinson and colleagues found using a similar passage scheme with an *Aeromonas* strain recently isolated from larval zebrafish digestive tracts [60]. Although Robinson et al. conducted their study with a zebrafish isolate, they too found a role for increased motility and extra-host transmission in improving host colonization. Given the unique ecological and evolutionary histories of wt MR-1 (no history with hosts) and the *Aeromonas* strain used by Robinson and colleagues (host isolate), the agreement of our results bolsters our collective findings and suggests that traits which elevate the transmissive capacity of host colonists can be crucial to the establishment of host associations. Furthermore, motility and chemotaxis have also been identified as significant colonization factors in other zebrafish isolates [61], and we observed elevated motility in the *Shewanella* zebrafish isolate we examined previously [7]. Additionally, these traits have been shown to alter biogeography along the length of the larval zebrafish intestine and promote persistence in the anterior portion of the zebrafish gut among bacterial symbionts native to zebrafish [62,63,64]. Together, these findings among native zebrafish symbionts suggest that traits linked with transmission can play a vital role in maintaining host associations, even when such associations may not strictly depend on direct contact with the host epithelium [63,64]. 

## 5. Conclusions

At the outset of our evolution experiment, we hypothesized that passaging a non-host-associated bacterium through the digestive tracts of larval zebrafish would enrich for genotypes whose adaptive effects would be realized within the host, and, given the substantial literature implicating pili in bacterial–epithelial interactions [11,12,20,21,22], we initially suspected that the MshL-T300P mutation would exert its adaptive benefit within the larval gut. However, we observed that dynamics taking place in the environment outside the host played a dominant role in determining which evolved genotypes successfully colonized hosts. Furthermore, although the MshL-T300P mutation did not impart a post-colonization adaptive advantage, it also did not preclude population growth within the gut. Since mutations accumulate as populations grow, additional passages could generate new mutations which improve survivorship within the gut relative to wt MR-1. In this way, over a longer evolutionary experiment, subsequent mutations might increase MR-1’s symbiotic affinity with this zebrafish host as the transition towards host association evolves. Ultimately, our observation that a non-host-associated bacterium could augment its ability to colonize a host by altering its behavior outside that host has important implications for future studies aimed at understanding which microbial traits enable bacteria to impact host biology. 

## Figures and Tables

**Figure 1 genes-12-00127-f001:**
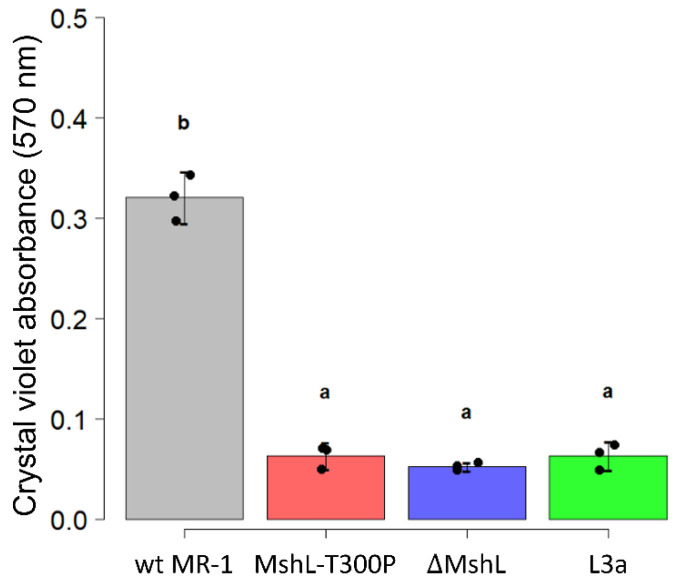
Crystal violet biofilm assay. Biofilm assessments were conducted in LCM. Absorbance at 570 nm corresponds to crystal violet staining intensity. Higher absorbance readings indicate more robust biofilms. Statistical groupings are indicated by letters above each box for a significance threshold of *p* < 0.05. Letters in common between groups indicate the absence of a significant difference between the groups’ means. MshL-T300P: missense mutation at MshL’s 300th amino acid position, ΔMshL: MshL deletion mutant, wt MR-1: unpassaged *S. oneidensis* strain, L3a: evolved isolate containing the MshL-T300P mutation.

**Figure 2 genes-12-00127-f002:**
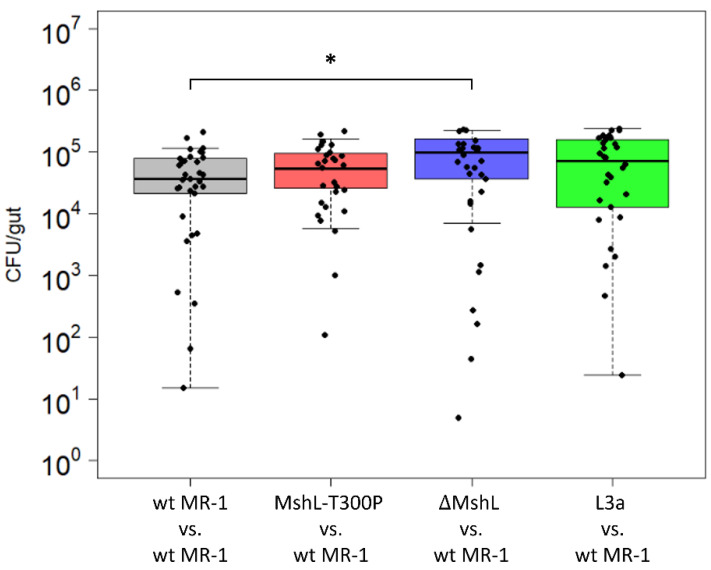
Colonization density achieved in larval guts after 72 h of colonization under competitive conditions. Dissected guts were plated on tryptic soy agar (TSA) and CFUs (colony forming units) were counted. Each point represents a single dissected gut (n = 30 larvae per comparison). Only the ΔMshL versus wt MR-1 results are different than the control group based on a two-tailed *t*-test with Bonferroni multiple comparison correction (*p* < 0.05: *). Three batches of larvae were used to generate these data, and batch was evaluated as a random effect.

**Figure 3 genes-12-00127-f003:**
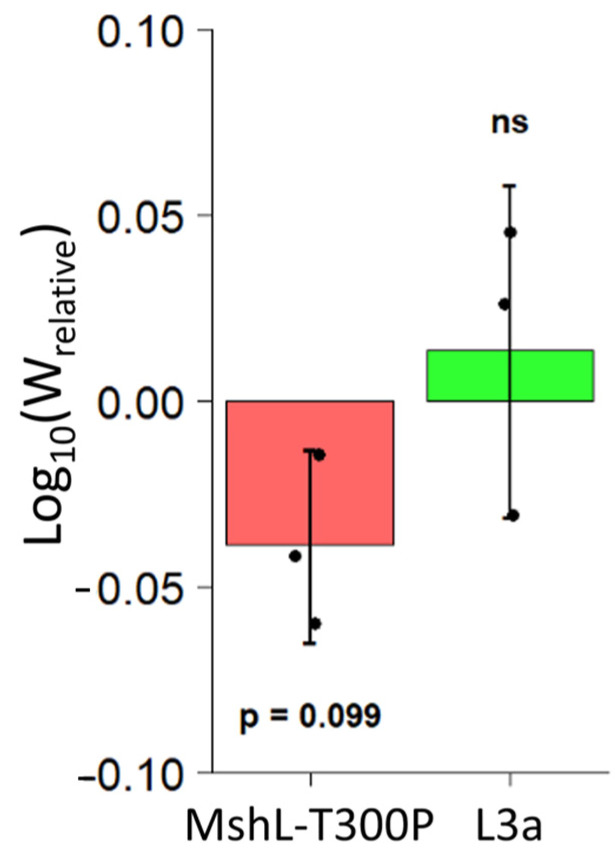
Competitive fitness of L3a and MshL-T300P mutant in vivo. Competitive in vivo growth rates were calculated (see methods) and then used to determine a relative in vivo fitness metric [32]. Each point represents the mean for a single competition flask where 10 colonized larvae were dissected at an initial time point and 10 larvae were dissected 10 h later. Each group of points was *t*-tested against the value zero, our expected value if there were no competitive advantage. No significant difference is indicated by “ns” (not significant).

**Figure 4 genes-12-00127-f004:**
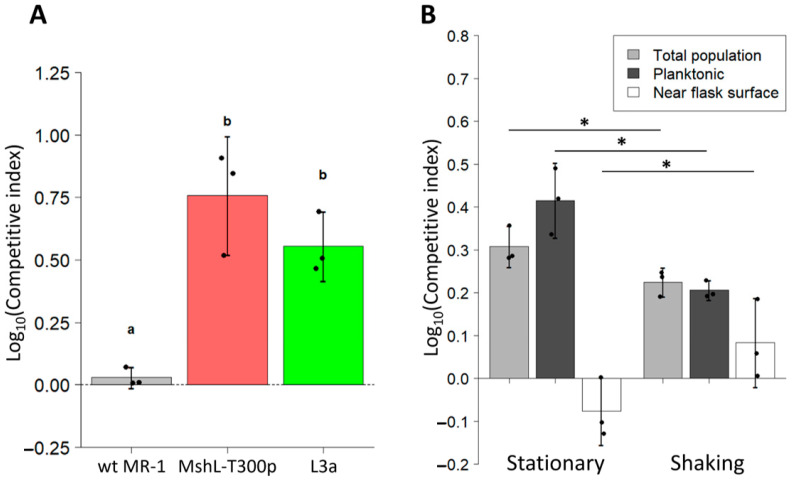
Competitive fitness of L3a and MshL-T300P mutant in larvae-conditioned media. (**A**) Competitive ability of L3a isolate and the MshL-T300P mutant against MR-1wt in LCM. Each point represents the competitive index measured for a single LCM competition flask. An ancestral competition against itself is shown as a control to represent the absence of a competitive advantage. Statistical groupings are indicated by letters above each box for a significance threshold of *p* < 0.05. Letters in common between groups indicate the absence of a significant difference in each group’s mean. The error bars indicate the standard error of the mean. (**B**) Competitive dynamics of the L3a isolate competing against its wt MR-1 ancestor in LCM under shaking and static conditions. Shown are the competitive indices which assess the performance of L3a relative to its wt MR-1 ancestor based on each competitor’s total population (light gray bars), planktonic population (dark gray bars), and flask-associated population (white bars). Error bars indicate the 95% confidence interval, and one-tailed *t*-tests were performed to assess the statistical differences between shaking and static conditions for the groups shown (*p* < 0.05: *).

**Figure 5 genes-12-00127-f005:**
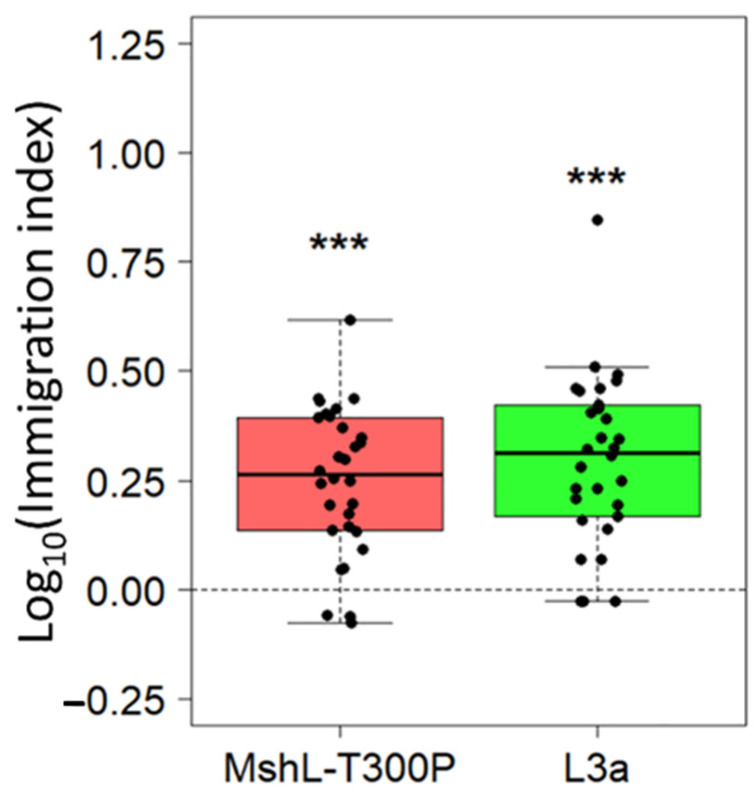
Relative per capita immigration of L3a and MshL-T300P mutants compared to wt MR-1. Each box shows the per capita immigration efficiency of the indicated strain relative to wt MR-1. Each point represents a single dissected and plated larval gut. Each group of points was *t*-tested against a mu value of zero, our expected value if there were no competitive advantage, and significance was determined after Bonferroni multiple-comparison correction (*p* < 0.001: ***).

## Data Availability

Refer to BioProject accession PRJNA633711 to access our genome assemblies and raw sequencing files (https://www.ncbi.nlm.nih.gov/bioproject/?term=PRJNA633711).

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
