# Peer review of "Msh Pilus Mutations Increase the Ability of a Free-Living Bacterium to Colonize a Piscine Host"

_genes, 2021, doi:10.3390/genes12020127_

Round 1

Reviewer 1 Report

The study by Lebov and Bohannan is a follow up of a previous study where the Authors obtained, by repeated passaging of a Shevanella free-living strain into the zebrafish gut, an evolved isolate with a missense mutation in the mshL pilus operon. This mutation confers increased swimming motility and decreased biofilm formation compared to the wild type. In this study the Authors investigated the selective advantage of such mutation in facilitating the onset of the host association in the gut. They do so through a series of experimental trials, comparing the original isolate L3a carrying the mutation (but also other mutations), and a constructed strain carrying only the missense mutation (MshL-T300P) with the wild type strain (the ancestor). By comparative sequence and protein structure analyses, they found that this particular mutation (MshL-T300P) has never been previously observed and it is predicted to alter the MshL stability, suggesting it is a loss-of-function mutation. Then the Authors actively seek to determine the selective advantage of this mutation compared to the wt.

Major results showed that 1) both mutants (L3a and MshL-T300P) showed higher competitive index in gut colonization after 72 hours, but lower biofilm formation with respect to the wt (the two mutants behave similarly); 2) the mutants do not show an increased carrying capacity; 3 ) the mutants do not outcompete the wt in vivo (all show similar growth rates in the larval gut); 4) the mutants outcompeted the wt in flasks with larva-conditioned medium (LCM); 5) The L3a strain is more planktonic and reached larger populations compared to the wt in LCM; 6) both mutants outcompeted the ancestor in a per capita immigration into the gut (one hour post-exposure of a mix of strains).

Overall, the mutants outperformed the wt in accessing the gut, but they do not show improved fitness within the gut.

I found this article interesting, the topic being central to the study of host-bacteria associations in the gut. The methods are well explained and there is a good analytical flow in the questions addressed and the way they are tackled.

I have however some concerns about the interpretation of some of the findings, especially on the adaptive role of this particular mutation in facilitating host colonization, which I think it would be worth some rethinking.

Below is a detailed review of all major and minor points that should be addressed.

  1. The Authors actively seek a selective advantage of the T300P mutation for the establishment of the host association. However, considering that this mutation was not previously observed in nature (T->P, 0,0% according to Table S1), the hypothesis to begin with is that this mutation might carry some kind of disadvantage on a long-term, where the increasing immigration phenotype observed (or improved swimming performance) might be just a side effect. Indeed, all experimental trials seem to confirm that this mutation does not have a fitness advantage in the gut. More importantly, the Authors did not demonstrate that the increase immigration rates of this mutant strain actually resolve into a long-term association (“establishment and maintenance”, as indicated at line 485). At line 85, “These results demonstrate that the initial steps during a microbe’s transition to host-association may be driven primarily by adaptations that alter its ability to access a host-based habitat, rather than adaptations that increase  fitness once a microbe has colonized a host.” The assumption here is that this mutation is truly adaptive to the gut, which was not demonstrated. The results actually point to the opposite direction, this mutation is not adaptive as there is no significant fitness advantage for this mutant once in the gut. I am puzzled by the fact that an increase swimming motility might favor host-colonization. This could be true in the sense that it increases host access, but motility per se is not expected to be a major advantage for settling into a host epithelium and form a stable community. Other observations also seem to reject this adaptive role. First of all, the host-associated strain of Shevanella (the Z12) apparently does not show this type of mutation, suggesting it is not required for colonization. Second, this mutation is not found in other strains, suggesting that it might carry some long-term deleterious effects. Third, if disruption of protein folding is a phenotype driving the early onset of the host-association, this should be observed in host-specific strains in general, which was not shown or discussed. It would be interesting to explore this scenario: are missense mutations in this operon significantly associated to host-specific strains? Finally, the Authors did not demonstrate that the increase immigration rates of this mutant strain actually resolve into a long-term association with the zebrafish gut (maximum observation time was 72 hours if I'm correct). Based on all this, I suggest toning down the statement that this specific mutation is relevant to the establishment of an association with the zebrafish gut.  The results are anyway interesting in relation to the phenotypic effect of this mutation, but I would suggest the Authors to limit the discussion to their observations.

  1. The additional LCM competitions testing for flask- surface- associated or planktonic state was performed only using the L3a strain. Why not testing the mutant? If the goal of this study is to understand the selective (dis)advantage of a specific mutation T300P, this experiment should have been performed also on the T300P mutant.

Minor comments:

  • Line 281. The Authors introduce the L3a evolved mutant. Could you please specify how this was obtained and in what it differs from the MshL-T300P mutant? This is essential information that is present in the previous study and mentioned in the Discussion of the present study, but that needs to be specified here at the beginning of the results.
  • Figure 3: L3 should be L3a
  • I found a bit confusing the use of ancestor, wild type, MR-1… I invite the Authors to stick with one name for the same strain in the text. If the ancestor is the wild type, I suggest to simply called it “(wt) MR-1” or “wt”.
  • Line 381: similar to our initial findings...” please indicate the reference. If this experiment was just repeated as part of a larger essay, this should be clearly stated, and previous work should be cited.
  • It would be useful to provide a description of predicted 3D structure of the mshl gene/operon at the beginning of the results, before discussing the role of this mutation in the pore structure.
  • Also Figure S1 A:  There are several enlarged residues, not just one. Please specify the structure of the pore.
  •  

Reviewer 2 Report

I read with great interest this study on mutant evolution in zebrafish populations. The authors performed experimental evolution experiments, but take ecological aspects into account. They addressed the functional mechanisms of the evolved strains for higher transmission and found that the mutation was associated with a loss of traits but higher motility of the bacterial strains (less biofilm formation). Density dependent processes outside the host additionally determined the evolutionary outcome of the process. This study is well performed and addresses nicely eco-evo aspects with respect to host-symbiotic interactions (focusing on bacterial funtional traits). I only have some suggestions for the revision, which you can find in the edited pdf.

Round 2

Reviewer 1 Report

I revised the latest version of the manuscript and I feel the Authors have adressed all my points, recognizing some of the limitations in the inferences made for this particularly mutant. The study is robust and well performed and I think it would provide a nice reference for future studies  on the genetics of host-bacteria interactions in the gut. I'm happy to support its publication.